# Combined Proteomics and Metabolism Analysis Unravels Prominent Roles of Antioxidant System in the Prevention of Alfalfa (*Medicago sativa* L.) against Salt Stress

**DOI:** 10.3390/ijms21030909

**Published:** 2020-01-30

**Authors:** Jikai Li, Jemaa Essemine, Chen Shang, Hailing Zhang, Xiaocen Zhu, Jialin Yu, Genyun Chen, Mingnan Qu, Dequan Sun

**Affiliations:** 1Institute of Grass Research, Heilongjiang Academy of Agricultural Sciences, Harbin 150080, China; ljk8699466@163.com (J.L.); hunkuncle@163.com (C.S.); cyszhanghailing@163.com (H.Z.); 2CAS Center for Excellence in Molecular Plant Sciences, Institute of Plant Physiology and Ecology, Shanghai Institutes for Biological Sciences, Chinese Academy of Sciences, Shanghai 200032, China; jemaa@picb.ac.cn (J.E.); chenggy@sibs.ac.cn (G.C.); 3Human Phenome Institute, Fudan University, Shanghai 200438, China; xc_zhu@fudan.edu.cn; 4Co-Innovation Center for Sustainable Forestry in Southern China, Nanjing Forestry University, Nanjing 210037, China; yu.jialin@yahoo.com

**Keywords:** salt stress, proteomics, alfalfa, metabolism, glutathione metabolism

## Abstract

Alfalfa is the most extensively cultivated forage legume worldwide, and salinity constitutes the main environmental scourge limiting its growth and productivity. To unravel the potential molecular mechanism involved in salt tolerance in alfalfa, we accomplished a combined analysis of parallel reaction monitoring-based proteomic technique and targeted metabolism. Based on proteomic analysis, salt stress induced 226 differentially abundant proteins (DAPs). Among them, 118 DAPs related to the antioxidant system, including glutathione metabolism and oxidation-reduction pathways, were significantly up-regulated. Data are available via ProteomeXchange with identifier PXD017166. Overall, 107 determined metabolites revealed that the tricarboxylic acid (TCA) cycle, especially the malate to oxaloacetate conversion step, was strongly stimulated by salt stress. This leads to an up-regulation by about 5 times the ratio of NADPH/NADP^+^, as well as about 3 to 5 times in the antioxidant enzymes activities, including those of catalase and peroxidase and proline contents. However, the expression levels of DAPs related to the Calvin–Benson–Bassham (CBB) cycle and photorespiration pathway were dramatically inhibited following salt treatment. Consistently, metabolic analysis showed that the metabolite amounts related to carbon assimilation and photorespiration decreased by about 40% after exposure to 200 mM NaCl for 14 d, leading ultimately to a reduction in net photosynthesis by around 30%. Our findings highlighted also the importance of the supplied extra reducing power, thanks to the TCA cycle, in the well-functioning of glutathione to remove and scavenge the reactive oxygen species (ROS) and mitigate subsequently the oxidative deleterious effect of salt on carbon metabolism including the CBB cycle.

## 1. Introduction

Alfalfa (*Medicago sativa* L.) is one of the most important feed beans in the world [1]. It is well known by its high biomass production and the nutritional value for livestock. About one-fifth of the world’s arable land and about half of the world’s irrigated land are affected by salinization [2]. In saline land, salts could be present in different forms, such as sodium chloride, magnesium and calcium sulfates and bicarbonates. These types of salts mostly exist in soil and water at various concentrations ranging from 50 to 600 mM for *Arabidopsis* [3] and from 125 to 500 mM for Brassicaceae and Solanaceae [4]. The mechanisms by which plants respond and/or tolerate salt stress are under intensive investigation [5,6]. In particular, the vast land areas of the world, such as China and the United States, possess arable land with high salt concentrations and severe drought/salt interactive conditions. Salt stress represents a major constraint among the environmental factors limiting the production of alfalfa [7].

Recent reports have shown that alfalfa tolerance to salt is often associated with changes in morphological and physiological characteristics, such as changes in plant structure and growth, changes in leaf epidermal thickness, stomatal regulations, germination and photosynthesis rate [8,9]. These changes are associated with a variety of cellular modifications, including changes in membrane and protein stability, increased antioxidant capacity and activation of hormonal signaling pathways, particularly those relying on “plant hormones” such as abscisic acid (ABA) [8]. However, the modulation in the above-mentioned changes at the cellular level constitutes a very complex biological response. Further detailed molecular profiling should be presented to fully comprehend the regulatory framework of plants to various abiotic stresses.

Currently, intensive and extensive studies are reported to focus on “Big Data-omics” tools encompassing metabolomics, transcriptomics and proteomics. A multi-omics analysis has emerged among the hottest buzzwords and as a promising approach in biological sciences, which could provide insights helping to enhance stress tolerance and be utilized in breeding and engineering programs aiming to create new lines or hybrid plants with improved desired agronomical traits [10]. Notably, multiple attempts have been made to obtain a profile of gene expression in alfalfa under saline conditions [11,12]. However, because of the limited correlation between transcript and protein levels, transcript profiles do not usually provide a complete descriptive story on the comprehensive biological pathways. Proteomics has become an important complement to mRNA data and an improved biological perspective on plant biology. 

Recently, several studies have sought to analyze and interpret the changes of protein expression in response to salt. Ma et al. have identified 2171 salt-responsive protein identities in 34 plant species, representing 561 unique proteins using a two-dimensional gel electrophoresis (2-DE)-based proteomic technique [13]. However, there are some limitations in this technique, including a large number of handled samples, limited reproducibility, and a smaller dynamic range than some other separation methods [10,14]. Furthermore, this technique is not automated for high-throughput analysis, and this represents another constraint to digitize and handle big raw data [15]. To date, parallel reaction monitoring (PRM)-based targeted mass spectrometry (MS), which is an ion monitoring technique, has emerged in the last decade as a powerful and valuable tool that is highly sensitive and reliable for the quantification of proteins [16]. 

Metabolism analysis is an effective and quantitative method used herein to elucidate the mechanisms of salinity tolerance. In the current study, we combined the PRM-based LC-MS/MS method and targeted metabolism to unravel the key proteins responsible for salt tolerance in alfalfa. Enriched biological pathways are comparable to biochemical analyses of enzymes and metabolites associated with antioxidant systems. The expression of differentially abundant proteins (DAPs) involved in the antioxidant pathway was further confirmed by its mRNA expression level. Finally, we summarize our current investigation by providing an easy model displaying the importance of the TCA cycle in alleviating the inhibitory effect of salt on the Calvin–Benson–Bassham (CBB) cycle and carbon metabolism through incorporation of reducing power and the reactive oxygen species (ROS) scavenging system. 

## 2. Results

### 2.1. Physiological Performance of Alfalfa Exposed to Salt Stress

Understanding the mechanism of tolerance in alfalfa to high NaCl concentrations of soil may certainly help to improve plant yields on saline lands. Herein, we considered two stressful factors for the plant: application of different salt concentrations for different periods of time (duration). Results showed that roots and shoots biomasses after 14 d exposure to salt gradually decreased with increasing salt stress treatment from 0 to 400 mM NaCl (Figure 1A). Foliar water deprivation and wilting were obviously observed for 400 mM NaCl treatment, where chlorophyll contents, ABS/RC (absorption per active reaction center), photosynthetic rates (Pn) and stomatal conductance (g_s_) decreased significantly relative to control without salt (0 mM NaCl). 

The original data for different salt concentrations and durations regarding these parameters are shown in Appendix A. Results of Table 1 demonstrate that salt stress treatments ranging from 0 to 400 mM exerted significant effects on all traits determined in this study, especially for proline content, Pn, g_s_ and chlorophyll content (Figure 1B; Table 1). In the same trend, duration of the same NaCl treatment for different days, ranging from 0 to 14 d, exhibited significant effects on 2 out of 9 traits measured (Table 1). The two traits significantly affected by salt treatment duration were chlorophyll content and catalase activity (CAT). In terms of interactive effects of NaCl magnitude and its duration, we found that the majority of traits were significantly affected, except for oxygen free radicals (ORF). 

### 2.2. Changes in H_2_O_2_ and Antioxidant Enzyme Activities 

To investigate whether oxidative stress is induced by salt stress for this forage legume alfalfa, we evaluated the activities of various antioxidant enzymes or metabolites, including ORF, peroxidase (POD), CAT and proline contents. Our findings demonstrated that proline was highly accumulated in salt-treated alfalfa with higher than 200 mM concentration, suggesting that plants triggered the ROS scavenging system to protect themselves from oxidative stress generated following excessive salt stress treatment. Activities of POD and CAT were significantly enhanced as well following salt stress treatments higher than 200 mM NaCl (Figure 2). The determined values showed that the oxygen free radicals (OFRs) were over-accumulated at NaCl concentrations higher than 100 mM relative to control. 

### 2.3. Identification and Functional Classification of DAPs

As depicted in Figure 1A, the salt stress concentration below 200 mM had no substantial effects on the alfalfa phenotype. For this reason, we selected to carry out the proteomics analysis at 200 mM NaCl. At this salt concentration, we predicted a considerable change regarding the molecular profiling in alfalfa leaves. We used tandem mass tag (TMT)-labeled proteomic techniques to analyze the differential abundance proteins (DAPs) induced by salinity effects. Peptide IonScore and mass error distribution are shown in Appendix A. Distribution of peptide length and count are depicted in Appendix A, while distribution of protein ratio and isoelectric point (pI) are displayed in Appendix A. Protein sequence coverage and molecular weight are shown in Figure 3A,B. The biological samples for either the control without salt or 200 mM NaCl treatments were clearly separated between treatments; however, biological replications of the same conditions were tightly close based on principal components analysis (PCA; Figure 3C). In total, we obtained 226 DAPs (*p* value < 0.05), based on a modified *t*-statistic with consideration of fold change >1.2 or <0.83 simultaneously shared between 108 down-regulated and 118 up-regulated DAPs (Figure 3D). The above evidence suggested that NaCl treatments could induce expression of large numbers of proteins, which allowed us to conduct a deep analysis on biological functions and molecular mechanisms. 

### 2.4. Gene Ontology (GO) and Kyoto Encyclopedia of Genes and Genomes (KEGG) Analysis

To further explore the gene function/biological pathways that are responsible for salt stress, we conducted gene ontology (GO) and Kyoto encyclopedia of genes and genomes (KEGG) analyses for DAPs induced by NaCl treatments. To interpret the major response, the three top biological pathways from enriched analysis were selected for further analysis. In terms of KEGG analysis, we found that proteins with significantly accumulated expressions were involved in three pathways of the biological process (oxidation-reduction process, hydrogen peroxide catabolic process and response to abscisic acid), three pathways of molecular function (glutathione transferase activity, heat-responsive proteins and electron transport oxido-reductase) and three pathways of cellular components (chloroplast, plastid stroma and chloroplast stroma) (Figure 4A). Conversely, proteins with significantly inhibited expressions belonged to several pathways, including three biological processes (ribosomal proteins, fatty acid and carbohydrate metabolism), three molecular functions (pyruvate dehydrogenase activity, metal ion binding and pyruvate kinase activity) and three cellular components (chloroplast, plastid and chloroplast parts) (Figure 4B). For other pathways except for the top three, complete and detailed descriptions are given in Appendix A.

In terms of GO analysis, we found that glutathione metabolism, metabolic pathways and amino sugar metabolism were significantly enriched in the list of NaCl treatment-induced up-regulated proteins (Figure 5A). In addition, carbon metabolism, pyruvate metabolism and fatty acid biosynthesis were significantly enriched in the list of down-regulated proteins induced by NaCl treatments (Figure 5B). 

Across these top three biological pathways that were significantly enriched following NaCl treatments, 34 DAPs were identified (Figure 6, Appendix A). Expression levels of genes in clusters I and III (classes I and III, Figure 6A,B) with up-regulation or down-regulation of DAPs (20 DAPs in total) were further confirmed by gene expression analysis (qPCR). The expression values for these 20 DAPs are presented in Appendix A. Our results showed that NaCl treatments induced about 4 times up-regulation across genes in cluster I (Figure 6B) including glutathione S-transferase1 (GST1), heat shock cognate 70 kDa protein (HSP70), NAD(P)-binding Rossman-fold protein (NADBR), monodehydroascorbate reductase (MODR), plastocyanin-like domain protein (POD), glutathione S-transferase (GSTa), glutathione S-transferase 3 (GST3), glutathione peroxidase (GLP), peptide-methionine (R)-S-oxide reductase (PMOR), ADP-ribosylation factor GTPase-activating protein (ADG10), aspartic protease in guard cell-like protein (APGC) and histone H2A (His2A). However, eight genes were down-regulated, such as legume lectin beta domain protein (LLBD), major intrinsic protein family transporter (MIP), 60S ribosomal protein (L3B), acyl-(acyl-carrier-protein) hydrolase (AcylH), methyltransferase (METR), 2-oxoisovalerate dehydrogenase E1 (2ODE1), alpha-L-arabinofuranosidase (ALDX) and ribulose bisphosphate carboxylase large chain (RbcL) (Figure 6A,B). 

Based on proteomics data, we deduced that salt stress could inhibit fatty acid metabolism and affect membrane integrity by decreasing the ratio of saturated and/or unsaturated lipid (DGDG and MGDG) impairment, as well as its intactness and fluidity, leading to ROS over-production (Figure 6C). Meanwhile, the oxidoreductases enzymatic activity involved in electron transport and glutathione metabolism were up-regulated (gene fold changes ranged from 3 to 5) (Figure 6C). This leads to overexpression of genes responsible for the ROS scavenging system. To enhance salt tolerance in alfalfa, cells accumulate proline as an osmoticum organic and ABA-inducible gene involved in stress signal transduction between the various organs of the plant (root and shoot) to help it relieve and withstand severe stress. 

As mentioned in Figure 5B, the protein amounts related to carbon metabolism were altered due to salt stress in alfalfa. To gain more insights about the mechanism of the metabolic pathway reprogramming under salt stress, we measured 107 targeted metabolites related to the Calvin–Benson–Bassham (CBB) cycle, photorespiration, tricarboxylic acid (TCA) cycle, glycolysis, starch synthesis, redox state and other derived metabolites from the TCA cycle using LC-MS/MS (Figure 7A). To determine the metabolites of the major six mentioned pathways (Figure 7A), five biological replicates were performed on either control or salt-treated alfalfa seedlings to 200 mM for 14 d. These samples were clearly clustered based on metabolites data between control and NaCl treatments with uneven percentages, 70.4% and 26.8% for PC1 and PC2, respectively (Figure 7B). We found 64 significantly abundant metabolites (DAMs) as a result of salt stress, distributed into 48 and 16 metabolites for down- and up-regulation, respectively, with all threshold *p* values < 0.05 (Figure 7C). Among these DAMs, 30 metabolites showed extremely significant differences between control and NaCl treatment (*p* < 0.01) (Figure 8A) and were involved in different metabolic pathways, including CBB and TCA cycles (Figure 8A). 

Based on metabolic data analysis, we expect that overproduction of ROS due to salt stress could certainly inhibit carbon metabolism not only by acting on the protein expression levels but also on the metabolic ones. On one hand, the number of metabolites related to carbon assimilation and photorespiration decreased by 40%, on average, as a result of salt stress, which led to around 30% reduction in Pn after 14 d exposure to 200 mM NaCl (Figure 1B; Figure 8B). On the other hand, accumulation of metabolites involved in the TCA cycle with about 5 times increase owing to salt stress effects could provide sufficient reducing power by poising the redox NADPH/NADP^+^ ratio in favor of NADPH accumulation (Figure 8B). The malate to OAA (Oxaloacetate) conversion step of the TCA cycle participates in providing extra reducing power (NADPH and/or NADH) through the accumulation of NADPH to help suppress and scavenge ROS, thus protecting carbon metabolism and the CBB cycle against inhibition. It seems that the extra reducing power provided by the TCA cycle is a prerequisite for the well-functioning of glutathione-dependent pathway reactions, under salt stress conditions, to remove ROS and thereby mitigate the inhibitory effects of NaCl stress on alfalfa. 

## 3. Discussion

Salt stress has become a serious threat to alfalfa growth and productivity. However, the mechanisms regulating salt tolerance and adaptation in alfalfa are less understood. In this study, through a combination of physiological, PRM-based proteomic techniques and targeted metabolic analysis, we performed a comprehensive analysis on key metabolic pathways directly implicated in salt stress responses and defense in alfalfa. These key metabolic pathways by salt stress are discussed as follows.

### 3.1. Glutathione Metabolism Plays a Crucial Role in Salt Tolerance

Glutathione (GSH; *γ*-glutamyl-cysteinyl-glycine) is a small intracellular thiol molecule existing simultaneously in reduced (GSH) and oxidized (GSSG) states and is considered as a strong non-enzymatic antioxidant [17]. It is known to regulate ROS metabolic functions [18] and acts among the ROS scavengers to improve tolerance to various abiotic stresses including salinity, drought, high and/or low temperature and toxic metals in maize [19], wheat [20], tobacco [21] and strawberry [22]. This is in line with our findings obtained for alfalfa under salt stress conditions (Figure 5A; Figure 7). Hence, the resulting salt resistance in alfalfa seems very likely to be largely caused by such metabolic regulation in the glutathione pool [21]. 

Accordingly, the increase in the glutathione pool helps to protect the cellular membranes and remove H_2_O_2_ by catalyzing glutathione oxidation (transition from the reduced to oxidized state of glutathione), thanks to the overproduction of NADPH or NADH, by malate to OAA conversion in the TCA cycle following salt stress treatment (Figure 8B). Thus, the TCA cycle operates as a safety valve in alfalfa under salt stress by supplying excess energy and reductant power necessary for the removal of ROS and the abolishment of its harmful effects on carbon metabolism, particularly on the CBB cycle, the main source of organic matter in the plant. Glutathione thereby prevents the oxidative denaturation of proteins under stress conditions by protecting their thiol groups. The denatured proteins serve as substrates for both glutathione peroxidase and glutathione S-transferase, which stimulate self-protection systems against NaCl-induced oxidative stress, as observed in alfalfa in the current study (Figure 6B). This protective role consequently triggers a number of antioxidant components such as CAT, POX and proline (Figure 2). These triggered enzymatic reactions have been reported to occur in many other species such as tobacco [23] and *Brassica* [24]. 

### 3.2. Mechanisms of Proline Stress Protection Coupling with ROS Scavengers

It is known that most abiotic stress such as salt stress generate oxidative stress through over-production of reactive oxygen species (ROS), consisting of singlet oxygen (^1^O_2_), superoxide free radical (O_2_^−^), hydrogen peroxide (H_2_O_2_) and hydroxyl radicals (OH^·^). These generated ROS forms cause oxidative damage that leads consequently to peroxidation of lipids, oxidation of proteins, inhibition of enzymes and damage to DNA/RNA. Proline is among the compatible solute compounds, and its accumulation is an indicator of excessive stressful conditions. The accumulation of proline helps plants to enhance their stress tolerance capacities and strengthen their antioxidant activity to detoxify the deleterious elements for the cell. 

The molecular mechanisms of how proline protects cells during stress are not fully and obviously understood. According to our data analysis, it seems that proline influences, thanks to its chemical properties, the redox systems such as the glutathione (GSH) pool (Figure 2). Because glutamate is a precursor of proline, GS activation may contribute to proline synthesis under salt stress [25]. The function of proline in stress adaptation is often explained by its property as an osmolyte (osomoticum organic) and its ability to balance water stress [26]. However, adverse environmental conditions mostly perturb intracellular redox homeostasis, requiring mechanisms that tend to relieve the oxidative stress repercussions on the cell. It has been also proposed that proline protective mechanisms are involved in the stabilization of proteins and antioxidant enzymes, immediate scavenging of ROS and balance of intracellular redox homeostasis (e.g., NADP^+^/NADPH and GSH/GSSG ratios), and also cellular signaling can be promoted by proline metabolism [26]. 

### 3.3. Inhibition of Photosynthetic Carbon Metabolism by Salt Stress

Photosynthesis efficiency (Pn) is a key indicator reflecting the physiological status of plants in response to abiotic stress. The decline in Pn due to stressful conditions, including salt stress, was observed in a number of studies [2,24,27]. The Pn may disclose the inhibitory effects of such stress conditions on the CBB metabolic pathways [2,27]. This mostly is due to a reduction in the activation state or temporary deactivation of the ribulose-1,5-bisphophate carboxylase/oxygenase (Rubisco), a primary site of the CBB cycle, as observed under high-temperature stress [28]. This inhibitory effect on the CBB cycle enzyme was also confirmed to happen following the reduction in RuBP activity or amount, the substrate of Rubisco, and it could be ascribable to the depressed expression of the Rubisco large subunit protein under salt stress (Figure 6A; Figure 8A). Notably, Rubisco is a bifunctional enzyme, in addition to RuBP carboxylation, which could also catalyze RuBP oxygenation, representing a reaction of the photorespiration pathway. 

We found that many metabolites involved in the photorespiratory pathway, such as glycolate, glyoxylate, serine and glycine, were dramatically inhibited by salt stress (Figure 8A). Coordination of the CBB and photorespiration cycles could be a reason explaining the inhibition of the photorespiratory pathway, as an intermediate glycerate-3-phosphate of the CBB cycle was produced by glycollate-2-phosphate and metabolized in the photorespiratory pathway. The reduction of this pathway might be also attributable to the over-reduction of the photosynthetic electron transport chain resulting from the over-accumulation of NADPH and/or NADH produced by the TCA cycle, thus leading to disequilibrium in cellular redox homeostasis under salt stress conditions [29]. 

### 3.4. Molecular Mechanism of ROS Signaling-Dependent Salt Tolerance

We provided a simplified model displaying the regulatory mechanism of ROS signaling pathways in salt tolerance in alfalfa (Figure 6C). Salt stress inhibits the carbohydrate metabolism pathway, pyruvate metabolism and Rubisco, thereby leading to a decline in the photosynthetic rates (Figure 1B; Figure 5B). At the cellular level, salt stress could induce over-accumulation of ROS such as OFRs [30]. The functional coordination of ROS-scavenging systems, such as the oxido-reductase enzymatic activities, enhances electron transport and glutathione metabolism from different cellular compartments. This could modulate and adjust the ROS level in the cells and prevent cellular damage, which may strengthen the salt tolerance capacity of plants. ROS signaling also encompasses antioxidant enzymes including POX and CAT. These enzymes could immediately consume and transform the harmful forms of ROS to other forms less detrimental for the cell through an appropriate scavenging process, or another way of detoxification such as immediate conversion of superoxide free radical (O_2_^.-^) to hydrogen peroxide (H_2_O_2_) by superoxide dismutase (SOD), then H_2_O_2_ to water by catalase (CAT). Furthermore, ABA is produced and subsequently causes the up-regulation of the expressions of some ABA-responsive genes. Altogether, increased proline contents, activation of ROS scavenging systems and enhanced ABA signaling pathways are responsible for the improved ability of alfalfa forage legumes to tolerate salt stress. 

## 4. Materials and Methods

### 4.1. Plant Material and Stress Treatments

To reveal the molecular response to salt tolerance in alfalfa, we selected a common cultivar of *Medicago sativa* L. (cv. Nongjing 1) in China. Seeds were germinated in the dark at 28 °C for 48 h and then transferred into 1/2 Hoagland nutrient solution as previously reported [10]. Then, the seedlings were independently placed in 0 (control), 50, 100, 200 and 400 mM NaCl 1/2 Hoagland nutrient solutions for 14 d, respectively. The experiment was carried out in a greenhouse with an average temperature of 26/18 °C day/night with a photoperiod 16/8 h dark/light and light intensity 200 μmol m^−2^ s^−1^. The air humidity was about 65%.

### 4.2. Physiological Measurements

Chlorophyll *a* fluorescence induction kinetics were measured at room temperature using a Plant Efficiency Analyzer (PEA, Hansatech, King’s Lynn, Northfolk, England). A 650 nm (peak wavelength) excitation light from three LED arrays was focused to the leaf surface. The light intensity of red light was 3000 μmol m^−2^ s^−1^, which was sufficient to induce the maximum fluorescence (F_m_). During recording, the sensor head received the fluorescent signal and digitized it in the control unit using a fast digitizer. An energy pipeline model was prepared using the Biolyzer HP3 software (chlorophyll fluorescence analysis program of the Bioenergetics Laboratory, University of Geneva, Switzerland). F_v_/F_m_ (variable fluorescence by maximum fluorescence) represents the maximum quantum yield for the PSII primary photochemistry. ABS/RC represents absorption flux per reaction center according to [31].

Same selected leaf sections were used for both chlorophyll fluorescence and gas exchange measurements with two independent experiments. We measured net photosynthesis (Pn), stomatal conductance (g_s_) and internal CO_2_ concentration (Ci) at PPFD (Photosynthetic Photon Flux Density) levels of 1500 μmol m^−2^ s^−1^. A portable photosynthesis system (LI-6400, LI-COR Biosciences Inc., Lincoln, NE, USA) was used at an ambient CO_2_ concentration (400 μmol.mol^−1^). The leaf temperature was maintained at 25 °C.

Absolute chlorophyll was extracted from homogenized seedlings in 8 mL of 80% (*v*/*v*) acetone in the dark at 4 °C for 72 h. The supernatant was quantified by monitoring the absorbance at 663 and 645 nm by a spectrophotometer [32]. The absolute chlorophyll content was calculated, as described by [33], according to the following formula: total chlorophyll (mg·l^−1^) = (8.02 × OD_663_) + (20.21 × OD_645_), where OD represents optical density. Chlorophyll content was expressed in mg g^−1^, fresh weight (mg g^−1^ fresh weight (FW)). 

### 4.3. POD, CAT and OFR Activity Analyses

To investigate the effects of salt stress on the ROS scavenging system, three antioxidant enzymes—POD (peroxidase), CAT (catalase) and oxygen free radicals (ORFs)—were selected for enzymatic assessment. Tissue extraction was prepared by homogenizing 1 g of leaf material in 4 mL of ice-cold 50 mM K-phosphate buffer (pH 7.0) containing 2 mM Na-EDTA and 1% (*w*/*v*) polyvinyl-polypyrrolidone (PVP). The homogenate was centrifuged at 10,000× *g* and 4 °C for 10 min. Crude tissue extracts were then stored at −80 °C or used immediately for subsequent POD and CAT assays. Anti-oxidase activities (including POD and CAT) were expressed in mg^−1^ protein. The activity of CAT was measured based on the decomposition of H_2_O_2_, and then the absorbance decreased at 240 nm by the method of [34]. 

### 4.4. Proline Determinations

The soluble proline content was assessed as reported by [35]. Cotyledons (200 mg) were ground in liquid nitrogen, and proline was extracted in 5 mL of 3% sulfosalicylic acid for 10 min. A mixture of acid ninhydrin and glacial acetic acid was added to the filtered extracts. The solution was kept at 100 °C for 40 min until the reaction was terminated in an ice bath. The reaction mixture was extracted with 5 mL of toluene. The absorbance of the toluene fraction was evaluated at 520 nm. The concentration of proline was estimated using l-valine as a standard and expressed in mg. g^−1^.

### 4.5. Metabolic Determinations

Trifoliate stage leaves were sampled from thirty-day-old alfalfa plants, either unstressed or exposed to 200 mM NaCl for 14 d (~2.5 mg for fresh weight), in 2 mL Eppendorf (Ep) tubes filled with pre-cooled metal beads and were immediately stored in liquid nitrogen. The samples were extracted with a ball mill at 30 Hz for 3 min. The extracted powder was dissolved with 1.3 mL methanol/chloroform and incubated at −20 °C for 4 h. The mixture was centrifuged at 2000 g at 4 °C for 10 min. The supernatant was removed to a new 2 mL Ep tube and stored at −20 °C for future use. To the remaining mixture of the original tube, 800 μL methanol was added and stored at −20 °C for 30 min, followed by gentle shaking for 3 min. The mixture was then centrifuged at 2000 g at 4 °C for 10 min. Combining the supernatant with the pre-extracted one formed 1.6 mL mixture altogether. The mixture was filtered with 0.43 μm organic phase medium (GE Healthcare, 6789-0404), and 200 μL was added to the HPLC sample bottle with a lined tube and store at −80 °C until use. The 77 metabolites with independent standard curves for each metabolite were determined using LC-MS/MS (AB Sciex Qtrap 6500; SCIEX). The column was a 100 × 2 mm Phenomenex Luna 3 μm NH_2_ (Catalog No.: 00D-4377-B0, 0.314 mL volume). Detection wavelength was 190 nm. Injection volume was 20 μL, and column temperature was 20 °C with 0.4 mL/min flow speed. Mobile phase: 15% buffer A (95% H_2_O and 5% ACN, pH = 9.5) + 85% buffer B (100% ACN), 0.2 mL.min^−1^. Five biological replicates were conducted for metabolic measurements. The linear regression index for standard curve for the determined 77 metabolites ranged between 0.88 and 0.99. The differentially expressed metabolites were determined by limma in R package. 

### 4.6. Protein Extraction

To further understand the global protein profiling of alfalfa exposed to salt stress, we performed proteomic analysis using leaf tissues subjected to 200 mM NaCl concentration for 14 d. The leaf total protein was extracted using cold acetone method as described previously with some modifications [36]. To improve the accuracy of the quantitative determination of the identified peptides, we performed tandem mass tag (TMT) labeling LC-MS/MS, coupled to a parallel proteomics monitoring (PRM)-based targeted proteomics approach. For protein extraction, ~5.0 g sample was homogenized with 1.5 g polyvinyl-pyrrolidone (PVP), floated with nitrogen liquid and ground to a fine powder in the mortar. Three biological replicates were conducted. The sample was then suspended in pre-cooled 10% TCA containing 90% acetone and 0.07% *β*-mercaptoethanol and stored at −20 °C overnight [37]. Thawed sample was centrifuged at 15,000 rpm for 30 min at 0–4 °C. Supernatant was discarded and pellet was eluted in pre-cooled 100% acetone containing 0.07% *β*-mercaptoethanol, followed by centrifugation at 15,000 rpm for 30 min at 4 °C. The precedent step was repeated approximately every 8 h until the supernatant became colorless. Then, the pellet was dried under vacuum infiltration to obtain a protein powder. The dried powder was dissolved in a lysis buffer containing 8 mol L^−1^ urea, 4% CHAPS (3-[(3-cholamido) dimethylamino]-1-propanesulfonate), 40 mmol L^−1^ Tris and 65 mmol L^−1^ DTT (dithiothreitol). The mixture was then homogenized using ultrasonic for 1 h, and centrifuged at 11,000 rpm for 30 min at 4 °C. Supernatants were collected and stored at −80 °C for the following proteomic analysis. The bovine serum albumin (BSA) assay was used as a standard for measuring protein concentration.

Protein digestion and peptide labeling were performed by Duplex TMT. Fifty micrograms of acetone lyophilized protein pellet of each sample was resuspended with 50 μL of 100 mM triethylammonium bicarbonate (TEAB). After the pellet was sufficiently suspended, 1.25 μL trypsin (about 1.25 μg) per 50 μg protein was supplied, and the sample was kept at 37 °C overnight as reported by [37]. A TMT double-stranded isotope labeling reagent set (90063) containing TMT 6-126 and TMT 6-127 labeling reagent vials was used, with 41 μL anhydrous acetonitrile. The reagent was dissolved for 5 min with occasional gentle shaking. Tubes containing samples were slightly centrifuged (1000× *g*, 1 min). TMT labeling reagent (41 μL) was carefully added to each 50 μg sample, and kept at room temperature for 1 h. Subsequently, we added 4 μL 5% hydroxylamine to each sample and incubated them for 15 min to stop the reactions. The samples were combined in equal amounts and then used for LC-MS/MS analysis as described in the following section.

Peptides were then analyzed on an AB SCIEX Triple time-of-flight (TOF) 5600 system and packaged with 5 μm C_18_ resin using a fused silica capillary emitter (inner diameter, 75 μm; length, 15 cm; New Objective, Woburn, MA, USA). For information-dependent acquisition (IDA), a survey scan obtained within 250 ms and 20 product ion scans were collected within 50 ms for each scan. The raw data file (wiff) obtained was analyzed on a mascot swissprot rice database using protein targeting software V.4.2 (AB SCIEX, Foster City, CA, USA). For both peptides and the MS/MS fragments, the threshold was specified to be ± 0.05 Dalton. Proteins were identified based on criteria that at least two peptides should match. The mass spectrometry proteomics data have been deposited to the ProteomeXchange Consortium via the PRIDE (PRoteomics IDEntifications database) [38] partner repository with the dataset identifier PXD017166.

### 4.7. Gene Ontology Information

Proteins identified by Mascot were screened by VennPlex software [39] to detect reproducible proteins. Significant proteins were classified as up- and down-regulated based on fold change parameter > 1.2 for the up-regulation and < 0.8 for the down-regulation; meanwhile, a *t*-test (*p* < 0.05) was considered. The differential Mascot protein ID was sought for at http://www.uniprot.org to detect the Uniprot ID. These Uniprot IDs are used in the National Center for Biotechnology Information (NCBI: http://www.ncbi.nlm.nih.gov/ protein/) to obtain protein sequences. The protein sequence in fasta format is mapped in the Rice Genome Annotation Project (RGAP: http://rice.plantbiology. msu.edu/analyses_searchblast. shtml). For functional analysis of the protein, the locus ID of the expression fold was used as input data in Mapman software version 3.6.0 RC1.

### 4.8. Quantitative Real-Time PCR

Total RNA from alfalfa seedlings was isolated using the Ultra-Pure RNA Kit (cwbiotech, Beijing, China) according to the manufacturer’s instructions. Complementary DNA (cDNA) was synthesized using the ExScript RT kit (Takara Bio. Beijing, China). Quantitative PCR (qPCR) was performed on a Realplex 4 Master Cycler using a gene-specific primer (Appendix A) using Super Real PreMix Plus with Syber Green 1 (TIANGEN Biotech., Beijing, China). Relative gene expression was assessed using a comparative cycle threshold method [40]. The relative gene expression was calculated as follows: 2^−ΔΔCT^ (ΔCT = CT, gene of interest^-CT^ versus actin) as described by [41], where the actin gene was used as a reference. A mixture cDNA of 30 leaf samples was used to perform the qPCR reactions to determine the primer efficiency as described previously [42]. All primer amplification efficiencies were between 99% and 104% (Appendix A). Three complete biological and technical replicates were performed.

### 4.9. Statistical Analysis

One-way ANOVA was used to compare the significant difference among NaCl treatments. We applied two-way ANOVA to analyze the interactive effects of two factors (salt concentrations and duration) on different physiological and biochemical parameters carried out in this study. A multi-omics data analysis tool, OmicsBean (http://www.omicsbean.cn), which integrated Gene Ontology (GO) enrichment and Kyoto Encyclopedia of Genes and Genomes (KEGG) pathway analysis, was employed to analyze the obtained DAPs and DAMs. A *p*-value < 0.05 (Fisher’s exact test) was used as the threshold to determine the significant enrichments of GO and KEGG pathways.

## 5. Conclusions

This present study reveals a PRM-based proteomic technique to dissect the potential molecular mechanism of alfalfa response to salt stress. In total, 226 DAPs were identified and divided into down-regulated (108) and up-regulated (118). Based on GO and KEGG analyses, we confirmed that salt stress enhanced antioxidant metabolism of glutathione and catalase to prevent the abolishment of carbon assimilation. This prevention seems to be offered by the TCA cycle through the generation of extra reducing power (NADPH), directly supplied to glutathione metabolism, to remove the ROS and release carbon metabolism, including the CBB cycle, from its deleterious, inhibitory effects. 

## Figures and Tables

**Figure 1 ijms-21-00909-f001:**
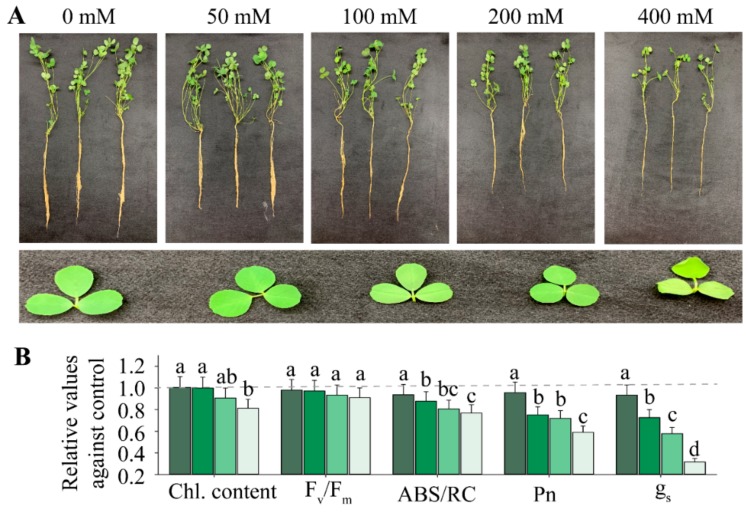
Performance of alfalfa seedlings exposed to different salt concentrations ranging from 0 to 400 mM for 14 d. (**A**) Plants and leaf images of alfalfa exposed to different NaCl concentrations as mentioned above the photographs for 14 d; (**B**) relative changes of different photosynthetic traits against control without salt (0 mM NaCl). One-way *ANOVA* was used to analyze the significant differences among different NaCl concentrations. Different alphabet letters represent significant differences at *p* < 0.05. Chl. content (chlorophyll content); F_v_/F_m_ (maximum fluorescence); ABS/RC (absorption per active reaction center); Photosynthetic rates (Pn) and stomatal conductance (g_s_). The gradient colors of bars from dark green to light green represent increasing NaCl concentrations from 50 to 400 mM. The original values for each photosynthetic trait induced by various NaCl concentrations and durations are summarized in Appendix A. Each bar data represents the mean (±SE) of 3 different replicates.

**Figure 2 ijms-21-00909-f002:**
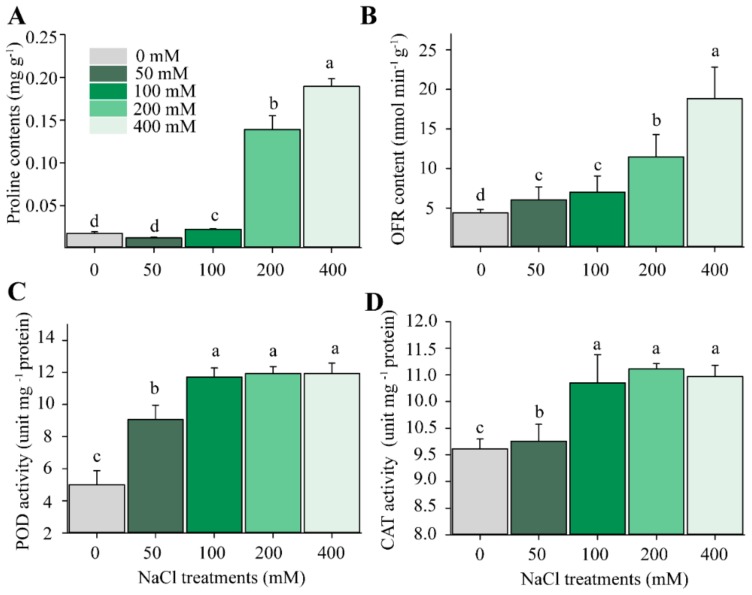
Effects of different salt concentrations on the proline content and activities of antioxidant enzymes in alfalfa. (**A**) Comparison of proline accumulation in alfalfa leaves enduring different NaCl concentrations for 14 d. (**B**–**D**) Comparison of contents of oxygen free radicals (OFRs), activities of peroxidase (POD) and activities of catalase (CAT) in alfalfa leaves treated with different NaCl concentrations for 14 d. One-way ANOVA was used to determine the significant differences among different NaCl concentrations. Each bar data represents the average (±SE) of 3 different replicates. Different alphabet letters represent significant differences at *p* < 0.05 based on one-way *ANOVA*.

**Figure 3 ijms-21-00909-f003:**
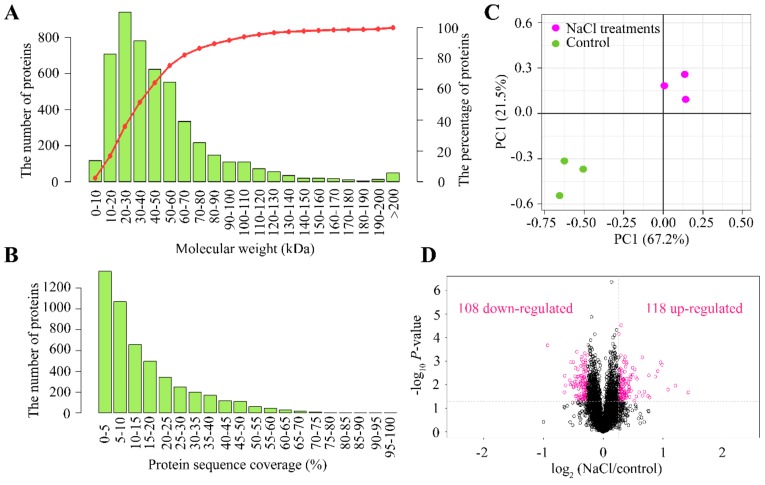
Protein coverage and differentially abundant proteins (DAPs) between NaCl treatments against control. (**A**) Numbers of proteins against different molecular weights. Y-axis on the right side represents the percentage of proteins and is indicated by the red line, while the Y-axis on the left side represents the number of proteins and is shown with green bars. (**B**) Number of proteins against different protein sequence coverages. (**C**) Principal component analysis (PCA) on three different biological samples for either salt-untreated (control) or treated with 200 mM NaCl treatment for 14 d (NaCl). NaCl treatments and control are depicted in pink and green colors, respectively. (**D**) Volcano plots representing the proteins with significant differences induced by 200 mM NaCl treatments for 14 d. Proteins with significant differences and large fold changes are shown in pink scatters.

**Figure 4 ijms-21-00909-f004:**
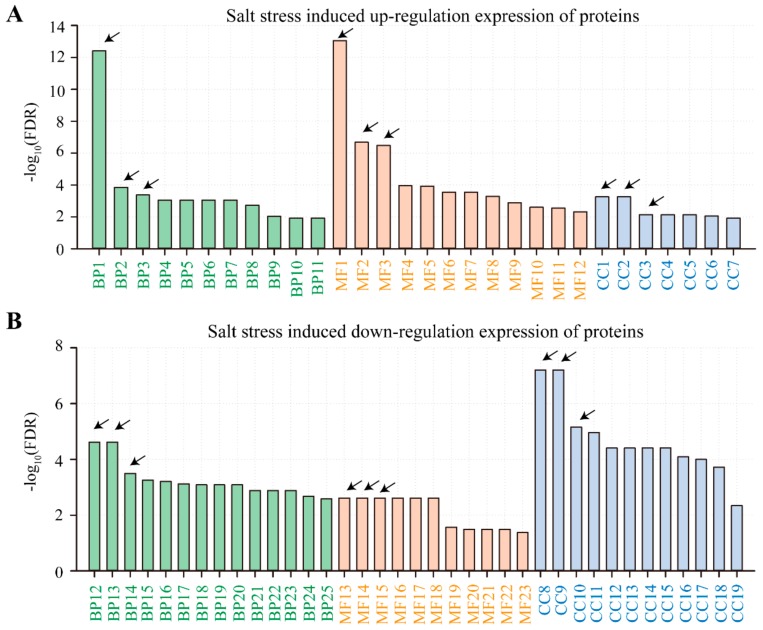
Kyoto encyclopedia of genes and genomes (KEGG) analysis on salt stress induced up-/down-regulation of expressed proteins. Panels (**A**,**B**) represent up-/down-regulation of proteins induced by 200 mM NaCl treatment in alfalfa seedlings for 14 d. Abbreviations: BP1-11 (biological process). MF1-12 (molecular function). CC1-7 (cellular components). The top 3 biological pathways were presented as follows for up-regulated proteins: (1) BP1 (biological process), oxidation-reduction process; BP2, hydrogen peroxide catabolic process; BP3, response to abscisic acid; (2) MF1 (molecular function), glutathione transferase activity; MF2, heat-responsive proteins; MF3, electron transporting oxidoreductase; (3) CC1-7 (cellular components)—chloroplast, plastid stroma, chloroplast stroma, cytoplasm, chloroplast part, plastid part and plastid. However, for down-regulated proteins of each pathway, we cited as well the top three: (1) BP12, ribosomal proteins; BP13, fatty acid; BP14, carbohydrate metabolism; (2) MF13, pyruvate dehydrogenase (acetyl-transferring) activity; MF14, metal ion binding; MF15, pyruvate kinase activity; (3) CC8, chloroplast; CC9, plastid; CC10, chloroplast part. Arrows represent top three pathways that in the list of different abundant genes in biological process, molecular function and cellular components. All the other up- and down-regulated proteins for each biological pathway are listed in Appendix A.

**Figure 5 ijms-21-00909-f005:**
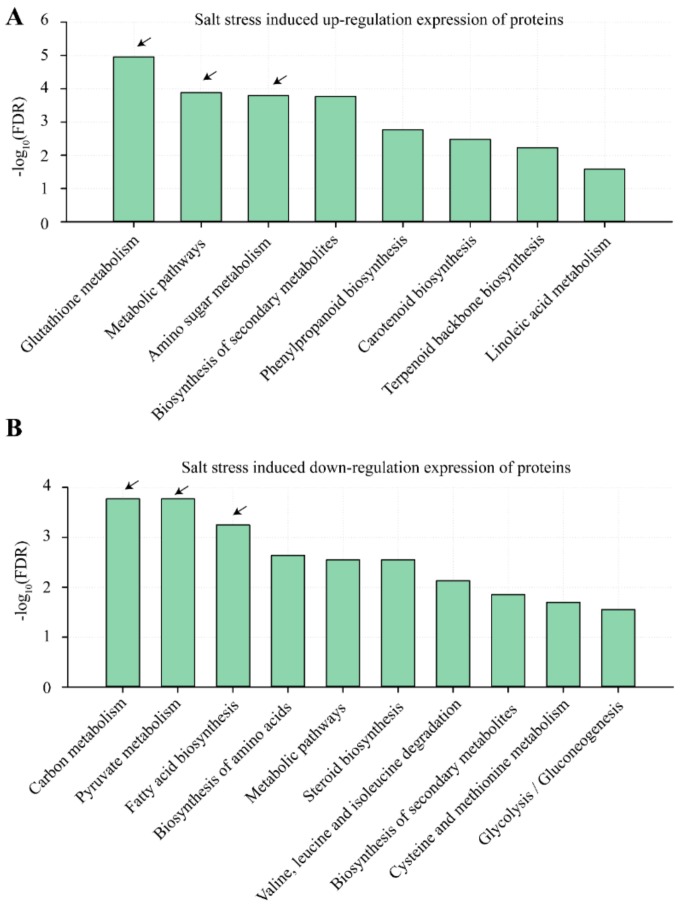
Gene ontology (GO) analysis on salt stress induced up-/down-regulation of expressed proteins. Panels (**A**,**B**) represent up-/down-regulation of proteins induced by 200 mM NaCl treatments in alfalfa seedlings during 14 d. The arrow represents the key biological pathways highlighted in this study. FDR means false discovery rate.

**Figure 6 ijms-21-00909-f006:**
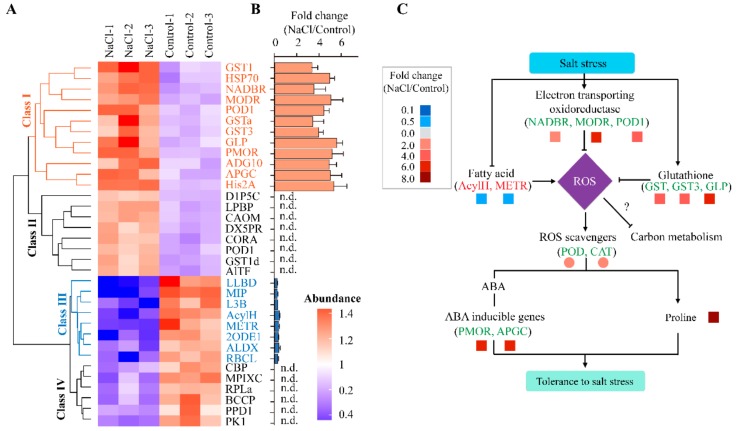
Global protein expression levels in alfalfa in response to salt stress. (**A**) Hierarchical clustering heatmap representing relative expression of the top 10% differentially abundant proteins (DAPs) between 200 mM NaCl treatment and control. Hierarchical clustering based on the abundant levels of protein identified was performed using Gene Cluster 3.0 software with the Euclidean distance similarity metric and linkage method. The resulting clusters were visualized using Java TreeView software. The fonts with blue color were further tested with qPCR. (**B**) Relative gene expression of DAPs determined by qPCR analysis. (**C**) A proposed model representing the regulation of the ROS scavenging system as a means of tolerance to salt stress. Symbol "?" represents unknown pathways. The fold change among these proteins in NaCl relative to control was depicted using different gradient colors. The expressions of values for different proteins were based on qPCR or physiological evaluations.

**Figure 7 ijms-21-00909-f007:**
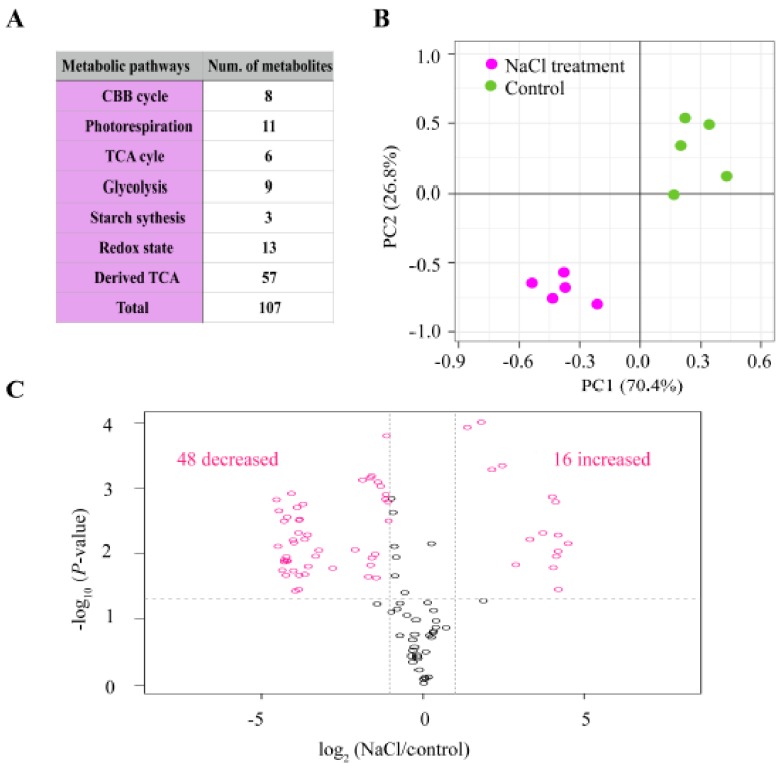
Statistical analysis of 107 targeted metabolites determined with LC-MS/MS in alfalfa seedlings enduring 200 mM NaCl for 14 d. (**A**) Number of metabolites determined for different metabolic pathways. (**B**) Principal component analysis (PCA) of five different biological replicates for either control without salt (control) or treated with 200 mM NaCl treatment for 14 d (NaCl). (**C**) Volcano plot representing the differentially abundant metabolites (DAMs) in alfalfa seedlings enduring the same salt concentration for the same duration (200 mM for 14 d).

**Figure 8 ijms-21-00909-f008:**
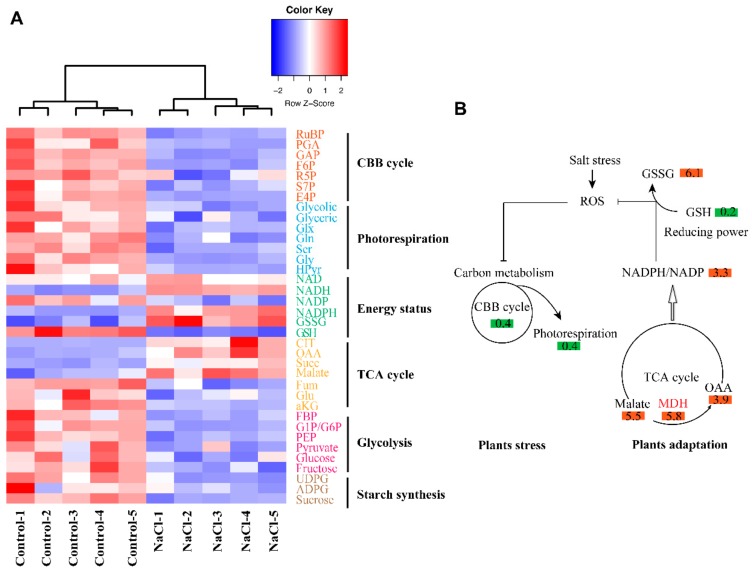
Metabolic reprogramming in alfalfa in response to salt stress. (**A**) Hierarchical clustering heatmap represents the metabolites related to different metabolic pathways in alfalfa seedlings enduring 200 mM NaCl for 14 d. The heatmap based on the identified protein abundance levels was created using Gene Cluster 3.0 software with the Euclidean distance similarity metric and linkage method. The resulting clusters were visualized using Java TreeView software. (**B**) A proposed model displays the triggering of ROS scavenging in alfalfa plants ascribed to the NADPH reducing power generated following conversion of malate to OAA (Oxaloacetate) in the TCA cycle as an adaptation strategy to salt stress. The numbers displayed in the boxes represent the ratio of metabolites under salt stress against control conditions (Salt/Ctrl), while green and red boxes represent down- and up-regulation of metabolic response in alfalfa to salt stress, respectively.

**Table 1 ijms-21-00909-t001:** Interactive effects of salt stress magnitude and its duration on physiological level and the activities of antioxidant enzymes in alfalfa leaves subjected to 200 mM NaCl for 14 d, via two-way *ANOVA*. Stars mean significance levels: *** *p* ≤ 0.001, ** *p* ≤ 0.01, * *p* ≤ 0.05.

Parameters	Duration	NaCl	Duration × NaCl
Chl. content (mg.g^−1^)	12.033 **	2.702 *	0.083 *
Proline content (mg.g^−1^)	2.168	47.614 ***	4.822 *
Oxygen free radical (OFR; mmol m^−2^ s^−1^)	0.192	0.147 *	0.01
Catalase activity (CAT; mmol m^−2^ s^−1^)	8.212 ***	1.856 *	3.207 *
Peroxidase (POD; mmol m^−2^ s^−1^)	0.247	0.388 *	6.563 *
F_v_/F_m_	0.008	1.583 *	0.073
ABS/RC	0.037	5.552 *	1.244 *
Pn (μmol m^−2^ s^−1^)	0.917	0.219 **	1.015 *
g_s_ (mmol m^−2^ s^−1^)	0.451	0.089 **	0.578 *

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
