# Peer review of "Combined Proteomics and Metabolism Analysis Unravels Prominent Roles of Antioxidant System in the Prevention of Alfalfa (Medicago sativa L.) against Salt Stress"

_ijms, 2020, doi:10.3390/ijms21030909_

Round 1

Reviewer 1 Report

Li et al. use a variety of omics techniques to determine the cellular pathways that are modified under drought stress. They compellingly identify a number of important pathways and provide an important contribution. I have only a few comments. 

Reference the range of salt concentrations and types of salt found in saline lands.

Figure 3A: Make clear which axis is for the bars and which is for the red line

Minor comments. The following phrases should be reworded

Line 51: "Besides, further detailed" just Further detailed

Line 63: "Recently, several studies are seeking" have sought?

Line 73: "used herein to well decipher and elucidate " just to decipher or to elucidate

Line 250: "we hence expect" change to we expect

Line 273: "are ambiguous and less understood" choose one or the other

Line 341: "Eventually, we provided " just We provided

Lines 320 - 323 Photosynthesis efficiency and physiological parameters are both followed by (Pn) to which does it refer? Perhaps remove the acronym?

Author Response

Li et al. use a variety of omics techniques to determine the cellular pathways that are modified under drought stress. They compellingly identify a number of important pathways and provide an important contribution. I have only a few comments.

Reference the range of salt concentrations and types of salt found in saline lands.

Revision done: answer added for this introduction lines 40 to 44.

“In saline land, salts could be present in different forms, such as sodium chloride, magnesium and calcium sulfates and bicarbonates. These types of salts mostly exist in soil and water at various concentrations ranging from 50 to 500 mM for Arabidopsis [3] and from 200 to 800 mM for Brassicaceae and Solanaceae [4].”

[3] Orsini, F.; D’Urzo, M.D.; Inan, S.; Serra, S.; Oh, D.; Mickelbart, M.V.; Consiglio, F.; Li, X.; Jeong, J.C; Yun, D.J; Bohnert, H.J; Bressan, R.A.; Maggio, A. A comparative study of salt tolerance parameters in 11 wild relatives of Arabidopsis thaliana. J Exp.Bot, 2010, 61, 3787–3798

[4] Popova, O.V.; Golldack, D. In the halotolerant Lobularia maritima (Brassicaceae) salt adaptation correlates with activation of the vacuolar H+-ATPase and the vacuolar Na+/H+ antiporter. J Plant Physiol. 2007, 164, 1278—1288

Figure 3A: Make clear which axis is for the bars and which is for the red line

Revision done: correction performed on Fig, 3A; lines 153-155.

“Y-axis on right side represents the percentage of proteins and indicated by red line, while Y-axis on the left side represents the numbers of proteins and shown with green bar.”

Minor comments: The following phrases should be reworded

Line 51: "Besides, further detailed" just Further detailed

Revision done: line 56, corrected to “Further detailed”

Line 63: "Recently, several studies are seeking" have sought?

Revision done: line 68, modified to “have sought”

Line 73: "used herein to well decipher and elucidate" just to decipher or to elucidate

Revision done: line 78, removed (well decipher) and kept to elucidate

Line 250: "we hence expect" change to we expect

Revision done: line 258, changed as recommended to “we expect”

Line 273: "are ambiguous and less understood" choose one or the other

Revision done: line 284, we removed "ambiguous and " and kept “are less understood”

Line 341: "Eventually, we provided" just We provided

Revision done: changed as recommended to “We provided”; line 352.

Lines 320 - 323 Photosynthesis efficiency and physiological parameters are both followed by (Pn) to which does it refer? Perhaps remove the acronym?

Revision done: to avoid confusion, we removed the physiological parameters and we kept “The Pn”; line 334.

Reviewer 2 Report

The article is well written and contains information's that deserve to be published after minor revision. I provide below a few suggestions that, if the authors decide to implement into the paper, the paper will improved.

Comments

Results: Some information that presented in Material and Methods is also presented in Results section. This information should be presented only in the Material and Methods.

Material and Methods

Lines 362-363: This phrase should be revised since it is not obvious if each salinity level is applied for 1,2,5,7 and 14 days.

References

Line 585: Medicago sativa should be written in italics.

Author Response

The article is well written and contains information's that deserve to be published after minor revision. I provide below a few suggestions that, if the authors decide to implement into the paper, the paper will improved.

Comments

Results: Some information that presented in Material and Methods is also presented in Results section. This information should be presented only in the Material and Methods.

Revision done: removed the sentence in results section: line 92-93; line 131-132.

We removed these sentences from the text:

“We applied two-way ANOVA to test the interactive effects of these two factors (salt concentrations and salt duration) on different physiological and biochemical parameters in alfalfa (Table 1). The original data for different salt concentrations and”; deletion starts from line 97.

“We sampled three fully expanded intact leaves from different 20-day old plants exposed for 14 additional days to 200 mM NaCl concentration. We used tandem mass tag”; deletion starts from line 138.

Material and Methods

Lines 362-363: This phrase should be revised since it is not obvious if each salinity level is applied for 1,2,5,7 and 14 days.

Revision done: The results presented in this study only used 14 days, so we removed other data description. Line 374, we removed the sentence “0, 1, 2, 5, 7 and” from this position “nutrient solutions for 0, 1, 2, 5, 7 and 14 days, respectively”

References

Line 585: Medicago sativa should be written in italics.

Revision done: line 602, “(Medicago sativa L.)” changed to “(Medicago sativa L.)”,

Reviewer 3 Report

The manuscript by Li et al. presents combined proteomic and metabolomic analysis of alfalfa exposed to salt stress to understand the potential molecular mechanism of salt tolerance in the species. Using established methods, the authors identified 226 differentially abundant proteins (DAPs) and determined 107 metabolites. GO-BP and KEGG pathway enrichment analysis suggested that the identified proteins were primarily involved in antioxidant system, including glutathione metabolism and oxidation reduction pathways. Several gene expression levels of the DAPs were also validated by qPCR. The experiments are appropriately designed and performed in a proper way. The dataset may include some important aspects. The reviewer is sure that such combined approaches can contribute to deeper understanding of molecular regulatory mechanisms and their metabolism in response to salt in alfalfa plants. However, I think that the treatment of data is shallow, and the manuscript does not provide sufficient information.

Methods and data interpretation

I feel that the findings are too preliminary, and the data analyses are quite shallow (see below). Treatment of the data does not go beyond GO and KEGG pathway enrichment analysis. Therefore, I think that the authors themselves fail to identify any novel findings, and hence the work, while comprehensive and comparative, does not greatly advance current knowledge in the area.

Identification of DAPs

You mentioned: L143-145 “In total, we obtained 226 DAPs (P value < 0.05) shared between 108 down-regulated and 118 up-regulated DAPs (Figure 3D).” This is confused for me. There was no information about the statistical tests. In addition, the authors described: L464-465 “Significant proteins are classified to up- and down-regulated, based on fold change parameter > 1.2 for the up-regulation and < 0.8 for the down-regulation.” This is too arbitrary. Why did you use t-statistic or linear models, such as limma (Smyth 2004)? Here, I would suggest that you should use modified t-statistic, because previous studies have demonstrated fold-change as well as the ordinary t-statistic was inferior to the modified t-statistic in terms of both reproducibility and accuracy. Because a typical transcriptome and proteome experiment contains a number of hypotheses and a few numbers of replicates, the authors should note high false-positive rates in identifying DAPs. See also, https://www.sciencedirect.com/science/article/pii/S2212968515000069. The authors should use the limma to identify DAMs in metabolite analysis.

PCA

Did you use any scaling or transformation methods?

GO/KEGG analysis

There was no description about GO/KEGG analysis. What kinds of statistical tools did you use? Did you use FDR correction for multiple testing?

Clustering methods in heatmaps

There was no description about your cluster analysis, such as metric/distance and linkage methods?

Data availability

If possible, please describe all the parameters for peptide annotation according to the standard guideline by HUPO-PSI. Furthermore, you should make your data available in a public database, such as PRIDE (EMBL-EBI, Cambridge, UK), PeptideAtlas (ISB, Seattle, WA, USA) (both of them are the founding members), MassIVE (UCSD, San Diego, CA, USA), jPOST (various institutions, Japan), iProx (National Center for Protein Sciences, Beijing, China), and Panorama Public (University of Washington, Seattle, WA, USA). Please also consider your metabolite data.

Minor comments/suggestions/errors:

Table 1 – Asterisks? Significant levels?

Fig. 3C and Fig. 7B – Please change the plot shape for readers.

Fig. 3 legend and Table S5 – “differentially expressed proteins (DEPs)” or DAPs? The same nomenclature should be used for both, otherwise it is a bit confusing.

Fig. 7C – Please use terms “decreased” and “increased”, instead of down- and up-regulated.

Fig. 6A and Fig. 8A – There was no description about your cluster analysis, such as linkage methods?

Conclusions – I think that this part is too poor.

Table S3 – Enriched?

Author Response

The manuscript by Li et al. presents combined proteomic and metabolomic analysis of alfalfa exposed to salt stress to understand the potential molecular mechanism of salt tolerance in the species. Using established methods, the authors identified 226 differentially abundant proteins (DAPs) and determined 107 metabolites. GO-BP and KEGG pathway enrichment analysis suggested that the identified proteins were primarily involved in antioxidant system, including glutathione metabolism and oxidation reduction pathways. Several gene expression levels of the DAPs were also validated by qPCR. The experiments are appropriately designed and performed in a proper way. The dataset may include some important aspects. The reviewer is sure that such combined approaches can contribute to deeper understanding of molecular regulatory mechanisms and their metabolism in response to salt in alfalfa plants. However, I think that the treatment of data is shallow, and the manuscript does not provide sufficient information.

Methods and data interpretation

I feel that the findings are too preliminary, and the data analyses are quite shallow (see below). Treatment of the data does not go beyond GO and KEGG pathway enrichment analysis. Therefore, I think that the authors themselves fail to identify any novel findings, and hence the work, while comprehensive and comparative, does not greatly advance current knowledge in the area.

Identification of DAPs

You mentioned: L143-145 “In total, we obtained 226 DAPs (P value < 0.05) shared between 108 down-regulated and 118 up-regulated DAPs (Figure 3D).” This is confused for me. There was no information about the statistical tests.

Revision done: More explanation is provided about DAPs identification as recommended.

Line 146, “based on modified t-statistic with consideration of fold change >1.2 or <0.83 simultaneously,”

In addition, the authors described: L464-465 “Significant proteins are classified to up- and down-regulated, based on fold change parameter > 1.2 for the up-regulation and < 0.8 for the down-regulation.” This is too arbitrary. Why did you use t-statistic or linear models, such as limma (Smyth 2004)? Here, I would suggest that you should use modified t-statistic, because previous studies have demonstrated fold-change as well as the ordinary t-statistic was inferior to the modified t-statistic in terms of both reproducibility and accuracy. Because a typical transcriptome and proteome experiment contains a number of hypotheses and a few numbers of replicates, the authors should note high false-positive rates in identifying DAPs. See also, https://www.sciencedirect.com/science/article/pii/S2212968515000069.

Revision done: we agreed using modified t-statistic, and indeed, we consider the fold change.

Revision added in lines 477-478 as follows: “meanwhile t-test (p < 0.05) was considered.”

The authors should use the limma to identify DAMs in metabolite analysis.

Revision done: sorry for inconvenience, we already used limma for DAMs analysis, just we didn’t mention in the text and actually we adjusted the issue according to your recommendation.

Modification added in line 434 “The differentially expressed metabolites were determined by limma in R-package.”

PCA

Did you use any scaling or transformation methods?

Revision done: We used auto scaling without transformation.

GO/KEGG analysis

There was no description about GO/KEGG analysis. What kinds of statistical tools did you use? Did you use FDR correction for multiple testing?

Revision done: we have added information about GO and KEGG analyses.  

Lines 498-502, “A multi-omics data analysis tool, OmicsBean (http://www.omicsbean.cn), which integrated Gene Ontology (GO) enrichment, Kyoto Encyclopedia of Genes and Genomes (KEGG) pathway analysis, was employed to analyze the obtained DAPs and DAMs. A p-value < 0.05 (Fisher’s exact test) was used as the threshold to determine the significant enrichments of GO and KEGG pathways.”

Clustering methods in heatmaps

There was no description about your cluster analysis, such as metric/distance and linkage methods?

Revision done: we have added information about cluster analysis in Figure 6 and Figure 8 legends.

Lines 272-276, “Hierarchical clustering heatmap represents the metabolites related to different metabolic pathways in alfalfa seedlings endured 200 mM NaCl for 14 days. The heatmap based on the identified protein abundance levels was performed using Gene Cluster 3.0 software with the Euclidean distance similarity metric and linkage method. The resulting clusters were visualized using Java TreeView software.”

Lines 227-232, “Global protein expression levels in alfalfa in response to salt stress. A, Hierarchical clustering heatmap representing relative expression of top 10% differentially abundant proteins (DAPs) between 200 mM NaCl treatment and control. Hierarchical clustering based on the abundant levels of protein identified was performed using Gene Cluster 3.0 software with the Euclidean distance similarity metric and linkage method. The resulting clusters were visualized using Java TreeView software.”

Data availability

If possible, please describe all the parameters for peptide annotation according to the standard guideline by HUPO-PSI.

Revision done: we have described all the parameter for peptide annotation in Table S4.

“Explanation of the parameters related to peptide annotation: Protein ID: Protein accession ID from fasta database. NaCl/control: ratio of protein expression levels under NaCl treatments over that under normal condition. Protein name: protein description from UniProtKB database. Gene_id and gene name: Gene accession ID from NCBI database. Coverage: The percentage of the protein sequence covered by identified peptides. Unique peptides: The numbers of peptide sequences unique to a protein group. Peptides: The number of distinct peptide sequences in the protein group. PSMs: peptide spectrum matches. The total number of identified peptide sequences for the protein, including those redundantly identified. AA: the total numbers of amino acids identified. MW[kDa]: molecular weight; calc.pl: Isoelectric point; Ratio(Salt/REF): Ratio of ion peak signal intensity in salt-treatment samples over that under normal condition.”

Furthermore, you should make your data available in a public database, such as PRIDE (EMBL-EBI, Cambridge, UK), PeptideAtlas (ISB, Seattle, WA, USA) (both of them are the founding members), MassIVE (UCSD, San Diego, CA, USA), jPOST (various institutions, Japan), iProx (National Center for Protein Sciences, Beijing, China), and Panorama Public (University of Washington, Seattle, WA, USA). Please also consider your metabolite data.

Revision done: we have submitted our proteomics data in PRIDE and we got response from the database team and recommended us to add some arguments to abstract and method the following sentences:

* For abstract:

The database stuff response:

We would recommend you to also include this information in a much abridged form into the abstract itself, e.g.

Sentences added "Data are available via ProteomeXchange with identifier PXD017166."; Lines 21-22.

* for method:

The database stuff response:

Please add to your manuscript the following sentence (typically in the "Methods" section or just before/in the Acknowledgements):

Sentences added: "The mass spectrometry proteomics data have been deposited to the ProteomeXchange Consortium via the PRIDE [24] partner repository with the dataset identifier PXD017166"; Lines 473-475.

[24] Perez-Riverol Y, Csordas A, Bai J, Bernal-Llinares M, Hewapathirana S, Kundu DJ, Inuganti A, Griss J, Mayer G, Eisenacher M, Pérez E, Uszkoreit J, Pfeuffer J, Sachsenberg T, Yilmaz S, Tiwary S, Cox J, Audain E, Walzer M, Jarnuczak AF, Ternent T, Brazma A, Vizcaíno JA (2019). The PRIDE database and related tools and resources in 2019: improving support for quantification data. Nucleic Acids Res 47(D1):D442-D450 (PubMed ID: 30395289).

Minor comments/suggestions/errors:

Table 1 – Asterisks? Significant levels?

Revision done: added

Line 117, we have added the sentence to table 1 legend: “Stars mean significance levels: *** p≤0.001, ** p≤0.01, *p≤0.05.”

Fig. 3C and Fig. 7B – Please change the plot shape for readers.

Revision done: thanks for your fruitful comments.

For PCA, yes we reshaped for readers. We used auto scaling, without log transformation.

New updated figures are inserted: Fig. 3C and Fig. 7B

Fig. 3 legend and Table S5 – “differentially expressed proteins (DEPs)” or DAPs? The same nomenclature should be used for both, otherwise it is a bit confusing.

Revision done: DAPs, corrected.

“Differentially expressed protein (DEPs)” changed to “differentially abundant proteins (DAPs) “

Fig. 7C – Please use terms “decreased” and “increased”, instead of down- and up-regulated.

Revision done: we have changed to decreased and increased in Fig. 7C as your recommended.  

Fig. 6A and Fig. 8A – There was no description about your cluster analysis, such as linkage methods?

Revision done: yes, we used “linkage methods” in line 231 (Fig. 6 legend) and in line 276 (Fig. 8 legend)

Conclusions – I think that this part is too poor.

Revision done: the conclusions were rewritten.

Table S3 – Enriched?

Revision done: we have corrected “Enriced” to “E nriched”.  

Round 2

Reviewer 3 Report

The manuscript has much improved.